# Validation of the Relationships Questionnaire (RQ) against the Experience of Close Relationship-Revised Questionnaire in a Clinical Psychiatric Sample

**DOI:** 10.3390/healthcare9091174

**Published:** 2021-09-07

**Authors:** Nahathai Wongpakaran, Justin DeMaranville, Tinakon Wongpakaran

**Affiliations:** 1Department of Psychiatry, Faculty of Medicine, Chiang Mai University, Chiang Mai 50200, Thailand; nahathai.wongpakaran@cmu.ac.th; 2Graduate School, Chiang Mai University, Chiang Mai 50200, Thailand; justinross_de@cmu.ac.th

**Keywords:** self-reporting measurement, attachment style, disagreement

## Abstract

Background: The Relationship Questionnaire (RQ) is a commonly used self-reporting questionnaire used to measure adult attachment styles. The RQ has two parts. RQ1, a single item where individuals have to indicate their preferred relationship style, and RQ2, where individuals can rate their relationship style in more detail using four different scales. Agreement is expected between the highest levels selected and the style chosen in RQ1. An advantage of the RQ is its brevity, whereas a disadvantage is that it constitutes a single item. A validation of RQ has not been clearly demonstrated, even though it has convergent validity in relation to other measurements in this area. Methods: 168 patients completed the RQ, the short version of the Experience in Close Relationships (Revised) questionnaire (ECR-R), and scales of depression and interpersonal problems. Regression analysis was conducted to examine the congruity in regard to attachment theory. Results: ratings from 15.5% of the patients showed disagreement between RQ1 and RQ2. Each type of attachment measured by the RQ was predicted by the ECR-R scores, as hypothesized. In the predictive analysis of depression and interpersonal problems, both RQ dimensions and ECR-R scores were coherent. Conclusions: RQ is a valid self-reported measurement that can be applied clinically on the condition that the rater identifies an agreement between RQ1 and RQ2.

## 1. Introduction

Attachment theory describes the origins of the patterns of human relationships. Genetic factors in early development interact with environmental factors (especially primary caregiver support) that lead to patterns of attachment behavior. Parental support during infancy initiates the development of the attachment system into a support-seeking response to distressing experiences. The parent is used as a safe haven and a secure zone around which the child explores their environment [1,2,3]. The attachment system is deactivated after the child feels supported and safe [4]. Receiving support when needed allows the attachment system to function optimally and enables a person to better maintain relationships and perform non-attachment-related activities [5].

Attachment types have evolved over time. Recently, clinicians and researchers have categorized attachment into four types: secure, fearful, preoccupied, and dismissing [6]. These types are based on a person’s model of the self and other. In addition, attachment is measured by dimensions, that is, anxiety and avoidance using continuous scores [7]. Figure 1 shows how those different concepts are incorporated.

Attachment has considerable psychosocial influence within medical research and health sciences. Attachment studies provide a predictive ability for clinically interesting outcomes. For example, insecure attachment types among post-natal women are associated with post-traumatic stress symptoms and the development of anxiety and depression [8,9]. Post-traumatic stress and low psychological adjustment were found in insecurely attached hospitalized burn patients [10]. Divorcees and separated individuals were more insecurely attached compared with married and remarried individuals [11]. Securely attached university students had higher self-esteem and lower shame compared with insecure students [12]. Amongst the elderly, depression significantly mediated the association between attachment anxiety and suicidal ideation [13].

Adult attachment can be measured via interviews, coding of the observed data, and the evaluation self-reported information [14]. Self-reported measures are commonly used to assess adult attachment in attachment research because of their convenience. At least 25 self-reported attachment questionnaires have been cited in the literature [14]. One of the two types of primary self-reported measures provides both categorical and dimensional data—such as in the Relationship Questionnaire (RQ) and the Attachment Style Questionnaire (ASQ) [6,15]—whereas the second type provides only dimensional data, such as in the Experience in Close Relationships (Revised) (ECR-R) questionnaire [7]. Dimensional data using continuous scores, such as in the ECR-R, are considered preferable and superior to categorical items as continuous analysis provides more statistical power. However, clinical interpretation may be difficult as a consensus on categorical attachment style scores is lacking.

Categorical measurements are used by clinicians because they are easy to communicate and explain in theoretical terms [8,9,10,16]. Some use it to maximize the comparability of the current long-term results with earlier outcomes [17]. Though categorical measures are convenient and fast to administer, respondents are forced to choose one particular style. This can lead to misclassification, especially for those whose scores place them near the boundaries of two attachment styles. Researchers using categorical measurements may want to delete contradictory data from individual respondents and use dimensional (continuous) scores for each attachment style (i.e., secure, preoccupied, fearful, and dismissing) to increase statistical power. For clinical use, however, there are not yet practical recommendations for clinicians [15].

It is evident that both ECR-R dimensional measurements and RQ categorical measures can predict outcomes of interest, such as depression and interpersonal problems. However, doubts may be raised about the results if both measures are not used. The ECR-R attachment anxiety and avoidance subscales can predict depression, depending on the studied population [17,18,19,20]. Some investigators have suggested using cluster analyses to categorize the avoidance and anxiety scores into four styles [21,22] for clinical applications, whereas for the categorical RQ it has been suggested to use dimensional data rather than nominal data. Categorical RQs are interesting for clinical use in the long-term due to the tool’s theoretical basis and convenience of use, but its validation has yet to be demonstrated explicitly.

It is important to validate the categorical RQ with the more reliable dimensional ECR-R to see how they correspond and can be used for clinical purposes. Based on the attachment measure (Figure 1), Fraley suggested that analyzing the two dimensions simultaneously in a regression framework allows for an interpretation of the results that is conceptually aligned with Bartholomew’s four attachment prototypes, i.e., secure (low anxiety and avoidance), preoccupied (high anxiety and low avoidance), fearful (high anxiety and avoidance), and dismissing (low anxiety and high avoidance) [23]. In addition to the agreement between these two measurements, a relationship between other outcome measurements should be observed. Related research has documented the association between attachment anxiety and attachment avoidance with depression and interpersonal problems. One way to prove the RQ’s validity is to illustrate the comparable and explainable results with ECR-R in association with depression and interpersonal problem outcomes.

The aims of this study were to examine the aforementioned criterion validity of the RQ against the ECR-R to see whether there was sufficient theoretical agreement between them, specifically among this sample with psychiatric disorders. The predictive ability for depression and some interpersonal problems was also assessed and compared between the two scales. We hoped to derive guidelines or suggestions on the use of RQ and ECR-R based on the results of this study.

## 2. Materials and Methods

This study design was cross-sectional.

### 2.1. Participants and Procedure

Participants included 168 patients who suffered from psychiatric disorders and who sought psychotherapy for additional treatment at the psychotherapy and personality disorders clinic at the Department of Psychiatry, Faculty of Medicine, Chiang Mai, Thailand. The patients came to the clinic via self-referral or a recommendation from their attending psychiatrists for treatment, ranging from combined therapy to regular medication therapy. Diagnosis based on DSM-IV-TR and DSM-5 was performed by the consultant psychiatrists. Psychodynamic psychotherapy was the main model for the patients. Patients who accepted psychotherapy completed the questionnaires for pretreatment therapy and pre- and post-session evaluations. ECR-R and RQ were completed only once in the pretreatment phase. All information provided was kept confidential, and personal information was kept confidential and undisclosed to any third party without the patients’ consent. This study was approved by the ethics committee of the Faculty of Medicine, Chiang Mai University.

### 2.2. Measurement

#### 2.2.1. Relationship Questionnaire (RQ)

The RQ has four measurable categories of attachment styles—secure, fearful, preoccupied, and dismissing [6]. The RQ is a single-item measure, consisting of four short paragraphs, each describing a prototypical attachment pattern as it applies to close relationships in adulthood. There are two parts, RQ1 and RQ2. In the first part, RQ1, participants were asked to select a paragraph-long description that best described them without providing a numerical rating. The essential statements for RQ1 are as follows. Secure attachment: “It is easy for me to become emotionally close to others. I am comfortable depending on them and having them depend on me. I don’t worry about being alone or having others not accept me”. Fearful attachment: “I am uncomfortable getting close to others, I want emotionally close relationships, but I find it difficult to trust others completely, or to depend on them. I worry that I will be hurt if I allow myself to become too close to others”. Preoccupied attachment: “I am uncomfortable getting close to others. I want emotionally close relationships, but I find it difficult to trust others completely, or to depend on them. I worry that I will be hurt if I allow myself to become too close to others”. Dismissive attachment: “I am comfortable without close emotional relationships. It is very important to me to feel independent and self-sufficient, and I prefer not to depend on others or have others depend on me”. In the second part, RQ2, participants are asked to rate their agreement with each prototype on a 7-point scale. The highest of the four attachment prototype ratings is then used to classify participants into an attachment category. RQ can make a model of the self (comparable to attachment anxiety in the ECR-R) and a model of the other (comparable to attachment avoidance in the ECR-R). As recommended by Bartholomew, the model of self is calculated by summing secure and dismissing items and then subtracting the sum of the preoccupied and fearful items, and the model of the other by summing the secure and preoccupied items and then subtracting the sum of the dismissing and fearful items [24]. Positive scores on these attachment representations indicate more positive models, and negative scores indicate more negative models (Figure 1).

Regarding psychometric properties, internal consistency cannot be calculated. Test-retest reliability can be estimated. The retest reliability for this measure was previously assessed as being in the range of 0.74–0.88 [25]. The construct validity of the RQ has previously been examined in [26], the authors of which conducted a cross-cultural study in 62 different cultures and found evidence for good convergent and discriminant validity of the RQ across cultures. RQ ratings have also shown good agreement with observer-based ratings of personality, self-reported ratings for interpersonal problems, and dimensional measures of attachment [21,27].

#### 2.2.2. The Short Version of the Revised Experience in Close Relationships Questionnaire (ECR-R-18)

This instrument was used to investigate close relationship experiences regarding attachment anxiety and attachment avoidance based on the assumptions of Brennan et al. [21]. According to the theory, the secure style corresponds to low anxiety and low avoidance, fearful to high anxiety and high avoidance, preoccupied to high anxiety and low avoidance, and dismissing to low anxiety and high avoidance. The ECR was later revised and validated by Fraley et al. [7]. It consists of 36 items: 18 for attachment anxiety and 18 for attachment avoidance. The response options range from 1—strongly disagree to 5—strongly agree. An example of attachment anxiety includes, “I need a lot of reassurance that I am loved by my partner” and, for attachment avoidance, “I try to avoid getting too close to my partner.” The Thai version of the ECR-R has demonstrated good validity and reliability [28]. This study used the short version of the ECR-R (ECR-R-18), containing 18 items: 9 for attachment anxiety and 9 for attachment avoidance. This short version demonstrated good validity and reliability among a nonclinical and clinical sample [29], and in the present study showed a Cronbach’s alpha value of 0.85.

#### 2.2.3. Depression Scale Measured Using the Outcome Inventory (OI-21)

The OI-21 is a scale used to measure common psychopathology in clinical practice, including depression, anxiety, somatization, and interpersonal difficulties. The OI-21 instructions directed respondents to respond to items based on how they felt over the past week. Response options were based on a 5-point Likert scale, i.e., values of 0 (never), 1 (rarely), 2 (sometimes), 3 (frequently), and 4 (almost always). Total scores were consistently interpreted—the higher the score, the higher the level of psychopathology. The OI-21 has been shown to have good validity and reliability [30]. The depression subscale was used in this study. In this study sample, the Cronbach’s alpha value of OI-21 was 0.79.

#### 2.2.4. Interpersonal Problems Measured Using the Inventory of Interpersonal Problems (IIP-32)

The IIP-32 was developed by Horowitz et al. [27]. It includes 32 questions, comprising 8 different interpersonal problems—domineering/controlling (DO), vindictive/self-centered (VI), cold/distant (CO), socially inhibited (SI), nonassertive (NO), overly accommodating (OA), self-sacrificing (SS), and intrusive/needy (IN). It uses a self-reported instrument in which respondents rate the severity of a wide range of interpersonal problems using a 5-point scale, ranging from 1 (not at all) to 5 (extremely). Each subscale has four items. Higher scores denote greater interpersonal difficulties. The IIP-32 demonstrated strong internal consistency (Cronbach’s alpha = 0.84) and acceptable test-retest reliability (intraclass correlation coefficient = 0.74). The Thai version demonstrated excellent reliability and validity [31]. The scale performed well with this study’s sample (Cronbach’s alpha = 0.88).

### 2.3. Statistical Analysis

Descriptive analysis, i.e., mean/standard deviation and percentage, was used for demographic data, i.e., age, sex, years of education, marital status, DSM-clinical disorders, and DSM-personality disorders. Discrepancies within the RQ were determined based on the agreement between RQ1 and RQ2, in that the scores in RQ2 should correspond with the style indicated in RQ1. If the corresponding style received the highest score, it was deemed agreeable; if not it would be deemed disagreeable. The percentage of agreement in each person was collected. To examine the criterion validity, we used ECR-R as a standard measure because it provides robust psychometric properties. Pearson/Spearman rank correlation and regression analysis were used for the investigation. Secure attachment would exhibit zero or low values of standardized coefficients for both attachment anxiety and attachment avoidance; preoccupied would exhibit positive coefficients for attachment anxiety but zero or low value in attachment avoidance; fearful would exhibit positive coefficients for both attachment anxiety and attachment avoidance; and dismissing would exhibit zero or low value of coefficients for attachment anxiety, but positive or high value of coefficients for attachment avoidance. Finally, a comparison of the predictive ability of both the RQ and ECR-R regarding mental health outcomes was carried out using linear regression.

## 3. Results

Table 1 shows the characteristics of all 168 patients. Most of them were female, with an average age of 31.8 years old, and diagnosed with MDD. The most common personality disorder was borderline PD.

Table 2 describes the distribution of attachment styles. Most of the respondents (79%) showed the insecure attachment style (fearful, preoccupied, and dismissing). Disagreement between RQ1 and RQ2 was observed in 26 out of 168 respondents (15.5%). The distribution of RQ2 based on RQ1 is shown below. It was expected that the score of the respective dimension would be significantly higher than the other dimensions, which all turned out as expected. To compare the differences between the scores of the respective style and the remaining styles, the independent *t*-test was used. The magnitude of the *t*-value would reflect how much each style conformed to the hypothesis. The higher the *t*-value, the higher the agreement would be between RQ1 and RQ2. Based on the magnitude of the *t*-value, the “fearful” item seems to be consistent between RQ1 and RQ2 (*t* = 17.983), compared to the dismissing prototype (*t* = 8.535).

The distribution of attachment style based on ECR-R score (using 4 as the cut-off) and RQ 2 scores. The difference between the two measures is most notably seen in secure attachment (Figure 2 and Figure 3).

As was expected, avoidance as measured by the ECR-R was significantly and positively correlated with the RQ dismissing item, but not, however, with the RQ preoccupied item (Table 3). The Greek version of ECR-R anxiety had a significant positive correlation with the RQ preoccupied item, but was not significantly correlated with the RQ dismissing item. Both ECR-R avoidance and anxiety were negatively correlated with the RQ secure item. Only ECR-R anxiety was positively correlated with the RQ fearful item, but not with ECR-R avoidance. Non-significant correlations between the ECR avoidance subscale and the RQ preoccupied item, as well as the ECR anxiety subscale and the RQ dismissing item, were expected according to our hypothesis for the RQ. As was expected, the model-of-self from the RQ items, as proposed by Bartholomew and Horowitz [6], was significantly positively correlated with ECR-R anxiety, and the model-of-other was significantly positively correlated with ECR-R avoidance, supporting the convergent validity of the ECR-R with respect to the RQ.

Table 4 shows the predicting coefficients of RQ styles by ECR-r using the whole sample, whereas Table 5 shows the predicting coefficients after the subjects with discrepancies in RQ were removed. A significant regression coefficient for the avoidance subscale for preoccupied attachment was observed (β = −0.203, *p* =0.008) compared to the original data (β = −0.134, *p* =0.091). In short, all relationships between the ECR-R and the RQ scores were as expected, given the current conceptualizations of the self-reported measurement of adult attachment [21,32]. Correlations between ECR-R anxiety and avoidance scores and OI depression were found, as follows—r = 0.53, *p* < 0.01, and r = 0.16, *p* = 0.062, respectively. The correlation coefficients between ECR-R anxiety scores and IIP subscales were 0.14, *p* > 0.05 for VI and 0.39, *p* < 0.01 for OA. The correlation coefficients between ECR-R avoidance scores and IIP subscales were between 0.03, *p* > 0.05 for SS and NI and 0.18, *p* < 0.05 for SI and OA (In Appendix A. Relationship between IIP subscales and ECR-R scalesn). Regarding the regression analysis, the results are shown in Table 6.

## 4. Discussion

According to our first objective of examining the agreement between the RQ and ECR-R, the results supported an agreement between the RQ and ECR-R, particularly when the cases containing disagreement between RQ1 and RQ2 were removed. Each attachment style described by RQ1 was correctly predicted by ECR-R anxiety and avoidance scores. As predictors, the accuracy of prediction of the RQ dimension (RQ2) depends largely on how much agreement exists between RQ1 and RQ2. For example, if one chooses the fearful option in RQ1, the RQ2 fearful score should be the highest, whereas the scores of secure, preoccupied, and dismissing should be much lower. Based on the results of the regression coefficients using ECR-R scores (Table 5), and *t*-statistic values (Figure 2), the RQ secure option seems to be the most consistent between the two questionnaires, whereas the RQ fearful option was the least consistent. It may be that fearful individuals not only scored highly on RQ2 fearful items but also scored highly on other RQ2 items, especially preoccupied items. This indicates that the RQ2 fearful statement may not help these clinical participants to differentiate between styles as expected. Evidence for this was shown in the significant relationship between RQ fearful and RQ preoccupied items (r = 0.19, *p* < 0.05). Notably, this inconsistency is usually found in clinical samples rather than in non-clinical samples. For example, in a study of Greek university students [33], it was revealed that the correlation between RQ styles and ECR-R anxiety and ECR-R avoidance subscales were as hypothesized. The RQ fearful item was significantly associated with both ECR-R anxiety and ECR-R avoidance. This contrasts with the clinical sample of the present study in that only ECR-R anxiety, but not ECR-R avoidance, was related to the RQ fearful item. Furthermore, the avoidance items had a lower magnitude of association with other variables compared to the anxiety subscale. Additionally, the avoidance items showed a relatively lower level of consistency compared to results found in anxiety items [28,29]. This contrast is sample-dependent. It depends on how the respondent reacted to those questionnaires, rather than reflecting a problem in the ECR-R avoidance category itself. Those clinical samples may be sensitive to ECR-R anxiety rather than avoidance, as compared with nonclinical samples. We believe that RQ-ECR consistency would be stronger in a healthy population that was undisturbed in terms of attention due to psychiatric symptoms.

The correlation between the self-model and ECR-R anxiety are consistent across studies, but not with the other model and ECR-R avoidance [22]. This could be due to the variations in attachment styles among the population. These findings have been endorsed by related psychiatric samples, in which the anxiety subscale plays a more important role than avoidance in relation to other psychological variables, e.g., depression [13,18].

In a prediction for some outcome variables, i.e., depression and cold and overly accommodating interpersonal styles, both ECR-R and the RQ supported each other. The outcomes were related more to ECR-R attachment anxiety, consistent with RQ fearful and preoccupied options, rendering confidence in the use of the RQ as a stand-alone tool. The strength of the correlation between attachment variables could determine which style would come into effect.

Although ECR-R scores provide more reliable data types for analysis, it is difficult to interpret the attachment style for each individual. It is helpful for clinicians or researchers to chart the distribution of attachment styles, as the RQ identified that nearly 79% were insecure, and mostly fearful. However, without validation with the ECR-R, we cannot be confident about the accuracy of the sample distribution or of the RQ’s predictive ability. The disagreement between the RQ1 and RQ2 reached 15.8%, implying the unreliability of the respondents. As suggested by Bartholomew [15], respondents who score incongruently between RQ1 and RQ2 should be cautiously approached. It is likely that they do not understand the items clearly due to impaired cognition or inattentiveness when completing the questionnaire. This often occurs amongst those with severe mental health problems such as depression. Bartholomew has suggested not to use such incongruent data for research, and our findings confirm that this is the case. Furthermore, an agreeable RQ can provide a reliable attachment style despite the fact that internal consistency cannot be obtained for a question with only one item. Without RQ1 as a confirmation, it may be difficult to ensure the credibility of the answer provided for the RQ2. However, the RQ can be used in clinical settings on the condition that there is good agreement between RQ1 and RQ2. A further question is what to do if the patient’s scores in the RQ are in disagreement. At what level of disagreement should the patient be asked to re-do the questionnaire once the discrepancy has been identified? Is the second-round RQ reliable and able to be used? These questions may not be easily answered and require further investigation.

### Limitations

The possibility of discordant responses between RQ1 and RQ2 remains a limitation of the tool. It may be that social desirability and faking behavior biases some participants to select contradictory attachment styles. This behavior is more likely in respondents with the insecure fearful and dismissing styles. There was evidence in the results of this study that insecure individuals gave somewhat unreliable answers. For example, some insecure individuals reported being secure in RQ1, but this was contradicted by RQ2. However, this theory of social desirability is not substantiated by the low total number of participants who identified as secure. It may also be those psychiatric symptoms, e.g., depression, which may cause cognitive impairment such as attention loss, could influence the disagreement between RQ1 and RQ2. Although this is possible, the correlation between RQ and ECR-R and other tools suggests that participant responses are reliable and that psychiatric symptoms cannot adequately explain the RQ discrepancies. Because the reasons for the discrepancy are unclear, we cannot now determine the exact type of attachment a discordant respondent may have used RQ alone. If RQ is to be administered alone, careful orientation and proper management should be provided in order to be certain that the respondent understands clearly and pays full attention when completing this questionnaire.

## 5. Conclusions

The Relationship Questionnaire (RQ) is short and easy to administer. It is a valid tool in assessing attachment styles, despite being a categorical measurement. As RQ items are descriptive, there may be discordance between RQ1 and RQ2 among some respondents who may not be attentive during the questionnaire. Practically, it can be used reliably in clinical settings on the condition that the RQ1 and RQ2 show good agreement.

## Figures and Tables

**Figure 1 healthcare-09-01174-f001:**
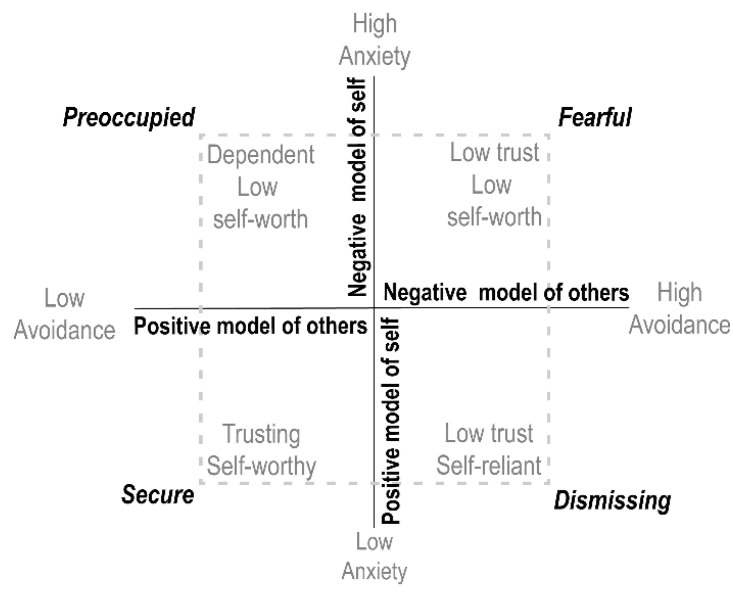
Four attachment styles based on two-dimensional approaches. This figure shows attachment styles based on two concepts. First, Brennan and colleagues’ two-dimensional components: the anxiety dimension (Y-axis) and avoidance dimension (X-axis) [21], and second, Bartholomew and Horowitz’s self/other model: self model (X-axis) and other model (Y-axis) [6]. The four quadrants produced by the two axes reflect secure, preoccupied, fearful, and dismissing attachment styles. For example, the preoccupied style is characterized by low avoidance and high anxiety according to the first concepts, and a positive model of the other and a negative model of self according to the second concept.

**Figure 2 healthcare-09-01174-f002:**
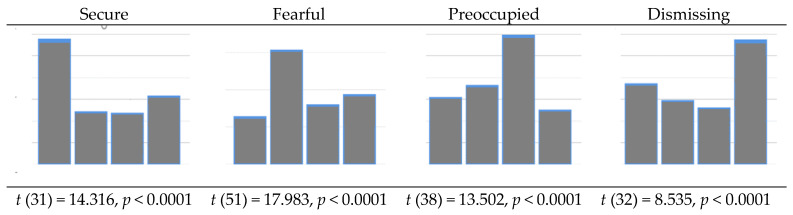
Discrepancy between RQ 2 scores in each attachment style.

**Figure 3 healthcare-09-01174-f003:**
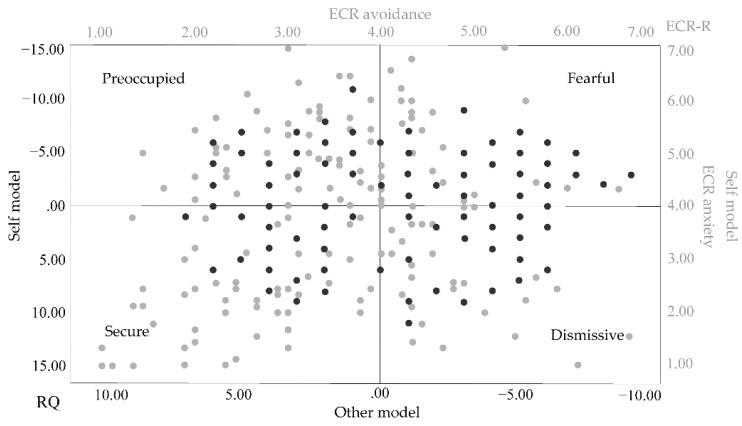
Distribution of attachment style based on ECR-R and RQ measures. Note: Gray dots represents cases identified based on ECR-R. Black dots represents cases identified based on RQ.

**Table 1 healthcare-09-01174-t001:** Demographic data and diagnoses based on DSM-5 (*n* = 168).

Variables	*n* (%)
Sex, female	106 (62.9)
Age	31.80 ± 14.4
**Education level**	
Elementary	6 (3.3)
High school	12 (7.1)
Bachelor’s degree	122 (73.1)
Master’s or higher	20 (11.8)
**Living status**	
Alone	122 (73.1)
With partner	46 (26.9)
**Clinical disorders**	
Major depressive disorders (MDDs)	80 (43.4)
Other depressive disorders	33 (19.8)
Bipolar disorder	12 (7.1)
Anxiety disorders	13 (7.5)
Others	30 (17.9)
**Personality disorders (DSM-5)**	
Borderline	43 (25.7)
Avoidant	42 (25.2)
Obsessive compulsive	16 (9.9)
Narcissistic	11 (6.9)
Antisocial	7 (4.0)
Dependent	6 (3.5)
Depressive	6 (3.5)
Histrionic	6 (3.5)
Schizotypal	6 (3.5)
Schizoid	3 (1.5)
none	22 (13.1)

**Table 2 healthcare-09-01174-t002:** Distribution of attachment style, mean and standard deviation of RQ and ECR-R scores.

Attachment Style	*n* (%)
RQ1: secure	33 (21.02)
RQ1: fearful	52 (33.12)
RQ1: preoccupied	39 (24.84)
RQ1: dismissing	33 (21.02)
RQ2: secure score, mean ± SD, median, min-max	3.57 ± 1.82, 4, 1–7
RQ2: fearful score, mean ± SD, median, min-max	4.08 ± 2.15, 4, 1–7
RQ2: preoccupied score, mean ± SD, median, min-max	3.58 ± 2.05, 4, 1–7
RQ2: dismissing score, mean ± SD, median, min-max	3.68 ± 1.80, 4, 1–7
Model of self (anxiety), mean ± SD, median, min, max	0.47 ± 4.69, 1, −11, 11
Model of other (avoidance), mean ± SD, median, min, max	0.62 ± 3.92, 1, −7, 9
ECR-R anxiety score, mean ± SD, median, min-max	3.68 ± 1.63, 3.78, 1–7
ECR-R avoidance score, mean ± SD, median, min-max	3.47 ± 1.22, 3.56, 1–7

SD = standard deviation, RQ = Relationship Questionnaire, ECR-R = the Revised Experience in Close Relationships questionnaire.

**Table 3 healthcare-09-01174-t003:** Comparison of RQ with the ECR-R scales.

	1	2	3	4	5	6	7	8
1 ECR-R anxiety	-							
2 ECR-R avoidance	0.238 **	-						
3 RQ2 secure	−0.205 *	−0.271 **	-					
4 RQ2 fearful	0.360 **	0.128	−0.293 **	-				
5 RQ2 preoccupied	0.441 **	−0.055	−0.142	0.188 *	-			
6 RQ2 dismissing	−0.160	0.288 **	0.112	0.028	−0.254 **	-		
7 Model of other	−0.008	0.330 **	−0.486 **	0.601 **	−0.456 **	0.523 **	-	
8 Model of self	0.479 **	0.029	−0.635 **	0.633 **	0.654 **	−0.522 **	0.073	-

RQ = Relationship Questionnaire, ECR-R = the Revised Experience in Close Relationships questionnaire; * *p* < 0.05, ** *p* < 0.01.

**Table 4 healthcare-09-01174-t004:** Predicting coefficients of RQ styles (categorical) by ECR-r (original data using the whole sample).

Variable	Unstandardized Coefficients		Standardized Coefficients	*t*	*p*-Value		
	B	Std. Error	Beta			R	Adj R Square
Secure							
attachment anxiety	−0.106	0.019	−0.411	−5.715	<0.0001	0.553	0.306
attachment avoidance	−0.109	0.026	−0.302	−4.208	<0.0001		
Fearful							
attachment anxiety	0.067	0.024	0.227	2.767	0.006	0.313	0.098
attachment avoidance	0.073	0.034	0.178	2.168	0.032		
Preoccupied							
attachment anxiety	0.113	0.021	0.416	5.289	<0.0001	0.413	0.17
attachment avoidance	−0.051	0.03	−0.134	−1.701	0.091		
Dismissing							
attachment anxiety	−0.071	0.02	−0.286	−3.504	0.001	0.326	0.106
attachment avoidance	0.076	0.028	0.219	2.684	0.008		

**Table 5 healthcare-09-01174-t005:** Predicting coefficients of RQ styles (categorical) by means of the ECR-R when subjects with discrepancies in RQ were removed.

Variable	Unstandardized Coefficients		Standardized Coefficients	*t*	*p*-Value		
	B	Std. Error	Beta			R	Adj R Square
Secure							
attachment anxiety	−0.121	0.018	−0.411	−5.715	<0.0001	0.634	0.393
attachment avoidance	−0.125	0.024	−0.302	−4.208	<0.0001		
Fearful							
attachment anxiety	0.068	0.023	0.241	2.915	0.004	0.335	0.112
attachment avoidance	0.067	0.030	0.183	2.216	0.028		
Preoccupied							
attachment anxiety	0.137	0.020	0.514	6.793	0.000	0.505	0.255
attachment avoidance	−0.070	0.026	−0.203	−2.679	0.008		
Dismissing							
attachment anxiety	−0.081	0.02	−0.332	−4.217	0.000	0.439	0.193
attachment avoidance	0.119	0.025	0.377	4.794	0.000		

**Table 6 healthcare-09-01174-t006:** Predictions for depression and interpersonal problems.

	Predictors	Unstandardized Coefficients		Standardized Coefficients	*t*	*p*-value
		B	Std. Error	Beta		
Depression						
ECR-R	anxiety	0.377	0.054	0.527	7.039	0.000
	avoidance	0.040	0.068	0.044	0.582	0.562
RQ2	Secure	−0.087	0.053	−0.135	−1.624	0.107
	Fearful	0.204	0.046	0.376	4.438	0.000
	Preoccupied	0.104	0.048	0.181	2.175	0.032
	Dismissing	0.002	0.053	0.003	0.041	0.967
Overly accommodating						
ECR-R	anxiety	0.915	0.173	0.421	5.291	0.000
	avoidance	0.331	0.222	0.119	1.494	0.138
RQ2	Secure	−0.395	0.169	−0.201	−2.339	0.021
	Fearful	0.364	0.146	0.220	2.501	0.014
	Preoccupied	0.367	0.156	0.206	2.352	0.020
	Dismissing	−0.178	0.172	−0.124	−1.034	0.303
Cold						
ECR-R	anxiety	0.715	0.203	0.293	3.526	0.001
	avoidance	0.533	0.259	0.171	2.059	0.041
RQ2	Secure	−0.691	0.164	−0.325	−4.208	0.000
	Fearful	0.676	0.142	0.375	4.767	0.000
	Preoccupied	0.098	0.153	0.051	0.641	0.523
	Dismissing	0.323	0.169	0.147	1.909	0.059

RQ = Relationship Questionnaire, ECR-R = the Revised Experience in Close Relationships questionnaire.

## Data Availability

The data is available upon request from the corresponding author.

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
