# Peer review of "Validation of the Relationships Questionnaire (RQ) against the Experience of Close Relationship-Revised Questionnaire in a Clinical Psychiatric Sample"

_healthcare, 2021, doi:10.3390/healthcare9091174_

Round 1
Reviewer 1 Report
This study aims to validate the Relationships Questionnaire (RQ) against the Experience of close relationship questionnaire-revised (ECR-R) because RQ is easier to administer in clinical settings. The authors found that, when RQ1 and RQ2 are consistent, the consistency with ECR-R is satisfactory. Thus, RQ can be used clinically when RQ1 and 2 agreement is met. I find the manuscript very well written, and the rationale of the study very clear. There are many good points in the General Discussion that I wish could be expanded/elaborated, which I outline below.
RQ1 seems to play a big role here in this study. Looking at the wording, I think the fearful statement is grammatically unclear (line 125 “but”), is this a translation error only?
Continuing from above, I find the 4 statements to be different in terms of conceptual structure. For example, the secure statement mentions “it is easy for me to…”, but neglects to talk about one’s desire. In other words, it may be easy for me to become emotionally close to others, but that does not necessarily mean that I WANT to. So in this sense, the secure statement only mentions one’s ability, which makes it more general than the preoccupied and fearful statements. Again, is this simple translation error, or does the design of these wording can potentially impact the results?
Was there a particular order in which the patients filled out the questionnaire/inventory? Order of the questionnaire sometimes may impact their subsequent answers.
The observation that only ECR-R anxiety, but not ECR-R avoidance, was related to the RQ fearful item is interesting. The authors seem to think this is due to the design of ECR-R avoidance statements (e.g., lower magnitude of association with other variables, lower level of consistency). But if this is true, then it remains to be explained why the Tsagarakis et al. (2007) study was able to find results in the authors’ hypothesized directions. Therefore, perhaps a better explanations would lie in the population (healthy vs. patient) and less so with the items (ECR-R avoidance)? If so, a deeper discussion that continues from Line 272 is called for.
Do the authors think this RQ-ECR consistency would be stronger, same, or weaker in the healthy population (who may not have a really obvious psychological trait like the patients do)? This would speak to the generalizability of the obserations from the present study, which would be suitable around line 272 as well.
Lastly, like the authors indicate on line 300, filling out these questionnaires/inventories is not only vulnerable to the risk of bias in social desirability, but it also requires a lot of self-insight from the participants toward themselves. Would this undermine the validity of these self-related questionnairse? The authors right now list this as an item in the Limitations section, but I would like to see the authors defend their positions (otherwise, why use questionnaires ever, at all?). One useful observation may be that the secure outcome in the present study is not very high, which may address the social desirability concern (since if desirability was true, secure should have the highest number). Another useful observation may be the mediation/correlation with depression and other symptoms, which suggest the patients do possess some kind of self-insight. These are just my suggestions, but my point is that there are lots of clues to address the abovementioned limitation, and it would be useful to write them out clearly in the Limitations section.
Minor comments:
Line 15: “closed” should be “close”
Line 291: “prove” should be “proof”
Author Response
1.RQ1 seems to play a big role here in this study. Looking at the wording, I think the fearful statement is grammatically unclear (line 125 “but”), is this a translation error only?
1.Response: Thank you for pointing this out. The original description is correct, but the concise one presented in the paper may be misleading. We have corrected this part keeping the complete original description for all styles.
2.Continuing from above, I find the 4 statements to be different in terms of conceptual structure. For example, the secure statement mentions “it is easy for me to…”, but neglects to talk about one’s desire. In other words, it may be easy for me to become emotionally close to others, but that does not necessarily mean that I WANT to. So in this sense, the secure statement only mentions one’s ability, which makes it more general than the preoccupied and fearful statements. Again, is this simple translation error, or does the design of these wording can potentially impact the results?
2.Response: As above, we have corrected this part keeping the complete original description for all styles.
3.Was there a particular order in which the patients filled out the questionnaire/inventory? Order of the questionnaire sometimes may impact their subsequent answers.
3.Response: Thank you for your thoughtful question. Yes, we were aware of that. To begin with RQ might have affected other following questionnaire items, e.g., interpersonal problems or mental health problems. Therefore, in administering the questionnaires, the participants were provided the order of questionnaires beginning with the questionnaires related to symptoms, followed by interpersonal problems, ECRR-18, and finally RQ.
4.The observation that only ECR-R anxiety, but not ECR-R avoidance, was related to the RQ fearful item is interesting. The authors seem to think this is due to the design of ECR-R avoidance statements (e.g., lower magnitude of association with other variables, lower level of consistency). But if this is true, then it remains to be explained why the Tsagarakis et al. (2007) study was able to find results in the authors’ hypothesized directions. Therefore, perhaps a better explanations would lie in the population (healthy vs. patient) and less so with the items (ECR-R avoidance)? If so, a deeper discussion that continues from Line 272 is called for.
4.Response: We totally agree with that. It does not have to do with the item but the population (i.e., clinical sample) that responded to those poorly compared with nonclinical samples found in Tsagarakis et al.’s (2007) study. We have cited the similar finding among depressed elderly that attachment anxiety played a greater role on the association with depression than attachment avoidance.
We have added some sentences to clarify this assumption as, “This contrast is sample-dependent. It depends on how the respondent reacted to those questionnaires rather than the problem of ECR-R avoidance itself. Those clinical samples may be sensitive to ECR-R anxiety rather than avoidance as compared with nonclinical samples.”
5.Do the authors think this RQ-ECR consistency would be stronger, same, or weaker in the healthy population (who may not have a really obvious psychological trait like the patients do)? This would speak to the generalizability of the obserations from the present study, which would be suitable around line 272 as well.
5.Response. Yes, we believe that RQ-ECR consistency would be stronger in a healthy population undisturbed by psychiatric symptoms as such patients. Healthy populations should have more self-insight and full state of attention that permit them better judgment in discriminating among each type of RQ. We have added this sentence to this discussion part as detailed below.
“This contrast is sample dependent. It depends on how the respondent reacted to those questionnaires rather than the problem of ECR-R avoidance itself. Those clinical samples may be sensitive to ECR-R anxiety rather than avoidance as compared with nonclinical samples. We believe that RQ-ECR consistency would be stronger in a healthy population undisturbed in attention due to psychiatric symptoms.”
6.Lastly, like the authors indicate on line 300, filling out these questionnaires/inventories is not only vulnerable to the risk of bias in social desirability, but it also requires a lot of self-insight from the participants toward themselves. Would this undermine the validity of these self-related questionnaires? The authors right now list this as an item in the Limitations section, but I would like to see the authors defend their positions (otherwise, why use questionnaires ever, at all?). One useful observation may be that the secure outcome in the present study is not very high, which may address the social desirability concern (since if desirability was true, secure should have the highest number). Another useful observation may be the mediation/correlation with depression and other symptoms, which suggest the patients do possess some kind of self-insight. These are just my suggestions, but my point is that there are lots of clues to address the abovementioned limitation, and it would be useful to write them out clearly in the Limitations section.
6.Response: Thank you for your keen observations. Let me respond to your interesting questions. Even though self-report has this limitation, we still need it in real world practices either for clinical or research practice. Interviewing for 1-2 hours to identify attachment style for each person is impractical. Like other self-report measurements, ECR-R and RQ may encounter reliability problems especially with a single type of questionnaire, e.g., RQ might be more vulnerable among participants with psychiatric symptoms. However, this effect may be mitigated using a multiple item questionnaire like ECR-R. Another concern regarding self-report is social desirability attitude (or faking good). This could occur among insecure individuals who may not want to be seen as insecure as characterized by RQ1. They may report themselves to be secure instead of insecure especially dismissing individuals (see Figure 3). Even though this could happen with dismissing individuals, not all will fake good. This is why secure is not highly rated. Additionally, for social desirability, some participants, experiencing psychiatric symptom, e.g., depression, may lose their full attention or experience some kind of cognitive impairment due to depressive state. They might be prone to endorse disagreement between RQ1 and RQ2. Because of these possible reasons, we will never determine what exact type of attachment that respective person would have when using RQ alone. We think that reliability should be high among secure people because they are reliable people based on the questionnaire but not for those who are insecure, especially fearful or dismissing. Evidence is shown from the results of this study that insecure individuals gave somewhat unreliable answers. For example, some insecure individuals reported to be secure by RQ1, contradicted the RQ2. If ECR-R and RQ were not used simultaneously, we would never determine what type of attachment that respective person would have. This is to confirm how RQ can be used in real practices or research based on our findings. If RQ is to be administered alone, it should be certain that the respondent understands clearly and pay full attention when completing this questionnaire. We have revised the limitation as described below.
Because all questionnaires used were self-report, they may have shown a risk of bias about social desirability especially among some dismissing individuals. We think that reliability should be high among secure people because they are reliable based on the questionnaire but not for those who are insecure, especially fearful or dismissing. Evidence is shown from the results of this study that insecure individuals gave somewhat unreliable answers. For example, some insecure individuals reported to be secure by RQ1, contradicted the RQ2. Further, regarding social desirability, some participants experiencing psychiatric symptom, e.g., depression, may lose their full attention or experience some kind of cognitive impairment due to depressive state. They might be prone to endorse disagreement between RQ1 and RQ2. Because of these possible reasons, we will never determine what exact type of attachment that respective person would have when using RQ alone. This is to confirm how RQ can be used in real practices or research based on our findings. If RQ is to be administered alone, careful orientation and proper management should be provided to be certain that the respondent understands clearly and pays full attention when completing this questionnaire.
7.Minor comments:
Line 15: “closed” should be “close”
Line 291: “prove” should be “proof”
7.Response
We have fixed all spots. proof” (now on line 311)

Reviewer 2 Report
It is an interesting aim, but you have to be very familiar with the RQ in particular with RQ1 and RQ2 to understand the approach. Even in the abstract the differentiation between RQ1 – a single Item where individuals had to indicate their preferred relationship style and RQ2 where individuals can rate on 4 different scales their relationship style in more details – should be explained shortly in the background – in (3) results suddently RQ1 and RQ2 results are presented without explanation.
P1 line33 – what is the meaning of “the attachment system will shut down after the child feels supported and safe”- the psychological, physiological, neural attachment system?
Figure 1 – the description refers to Hazan’s two-dimensional components – without reference – Bartholomey’s self- and other-model – without reference.
The authors indicate that the study is cross sectional – but on page 4 it is indicated that patients (how many?) who accepted psychotherapy completed the questionnaire for pretreatment therapy and post-session evaluation. Which one of the measurements are used in the study the pre- or post-psychotherapy ones – or if both, it is not a cross sectional study anymore.
P6 it is interesting that 21% describe themselves as secure - in comparison with a non-clinical sample is it high?
P7 t-tests are indicated – between which scores? Please explain in more details.
Figure 3 – which dots are related to RQ which ones to ECR-R?
Table 3 - you give numbers to the ECR-R scales but don’t use these numbers for the description of the scales on page 8. What is the meaning og G-ECR-R
Table 5 – the heading of the attachment style is located at the left side for secure but for the other ones in the middle – in the tabeles you name the Beta-values as Beta, in the text as b not as ß
P9 – it might be better understandable, if you provide a table of the correlations between the scales
Discussion: reference 34 is missing,
P10 line 285 – reference is missing
References – are presented in different citation styles - it should be consistently cited in one style
Author Response
Reviewer 2
8.It is an interesting aim, but you have to be very familiar with the RQ in particular with RQ1 and RQ2 to understand the approach. Even in the abstract the differentiation between RQ1 – a single Item where individuals had to indicate their preferred relationship style and RQ2 where individuals can rate on 4 different scales their relationship style in more details – should be explained shortly in the background – in (3) results suddently RQ1 and RQ2 results are presented without explanation.
8.Response. We appreciate this suggestion. We have revised as shown below.
Background: Relationship Questionnaire (RQ) is a commonly used self-reporting questionnaire to measure adult attachment styles. RQ has two parts. RQ1 – a single item where individuals had to indicate their preferred relationship style and RQ2 where individuals can rate their relationship style in more details using 4 different scales. The agreement is expected to be the highest level in the style chosen in RQ1.The advantage of the RQ is its brevity, whereas its disadvantage is that it constitutes a single item. A validation of RQ has not been clearly demonstrated even though it has convergent validity to other measurements; (2) Methods: 168 patients completed the RQ and the short version of experience of close relationship questionnaire-revised (ECR-R), the Depression and interpersonal problems scales. Regression analysis was conducted to examine congruity regarding attachment theory; (3) Results: Ratings from 15.5% of the patients disagreed between RQ1 and RQ2. Each type of attachment measured by the RQ was predicted by the ECR-R scores as hypothesized. In predictive analysis of depression and interpersonal problems, both RQ dimensions and ECR-R score were coherent.; (4) Conclusions: RQ is a valid self-report measurement that can be applied clinically on the condition that the rater provides an agreement between RQ1 and RQ2.
9.P1 line33 – what is the meaning of “the attachment system will shut down after the child feels supported and safe”- the psychological, physiological, neural attachment system?
9.Response: Hazan and Shaver explained the attachment system is similar to physiological systems. We have added some sentences for more clarification as shown below.
“Parental support during infancy initiates the development of the attachment system into a support seeking response to distressing experiences. The parent is used as a safe haven and a secure zone around which the child will explore the environment [1-3]. The attachment system will deactivate after the child feels supported and safe [4]. Receiving support when needed allows the attachment system to optimally function and enables a person to better maintain relationships and perform non-attachment related activities [5].”
10.Figure 1 – the description refers to Hazan’s two-dimensional components – without reference – Bartholomey’s self- and other-model – without reference.
10.Response: We have added the reference. We also corrected the wrong citation of Hazan, and replaced with ‘Brennan and colleagues’.
11.The authors indicate that the study is cross sectional – but on page 4 it is indicated that patients (how many?) who accepted psychotherapy completed the questionnaire for pretreatment therapy and post-session evaluation. Which one of the measurements are used in the study the pre- or post-psychotherapy ones – or if both, it is not a cross sectional study anymore.
11.Response: As ECR-R, RQ, and other measures were assessed only once at pre-psychotherapy, we consider it a cross-sectional design. We have revised this part for more clarification as “ECR-R and RQ were completed only once at pretreatment phase”
12.P6 it is interesting that 21% describe themselves as secure - in comparison with a non-clinical sample is it high?
12.Response: We think that 21% is not that high compared with other insecure attachments. Non-clinical samples usually report higher rates of secure attachment than other types of insecure attachment, for example, those of the studies of Ostacoli and Civilotti (Ostacoli et al., 2020) (Civilotti, Dennis, Acquadro Maran, & Margola, 2021). However, we believe that the number should have been lower than 21% if social desirability was not the case.
13.P7 t-tests are indicated – between which scores? Please explain in more details.
13.Response: We explain this point more by adding the following statements, “To compare the difference between the scores of the respective style and the remaining styles, independent t-test was analyzed. The magnitude of t-value would reflect how much each style conformed to the hypothesis. The higher the t-value, the higher the agreement would be between RQ1 and RQ2.”
14.Figure 3 – which dots are related to RQ which ones to ECR-R?
14.Response: we have added this information in the legend.
Grey dot represents a case identified based on ECR-R. Black dot represents a case identified based on RQ.
15.Table 3 - you give numbers to the ECR-R scales but don’t use these numbers for the description of the scales on page 8. What is the meaning og G-ECR-R
15.Response The number for each variable (horizontally) corresponds to the numbers in the vertical plane. It has been used only to help the reader to see clearly about each pair of variables. An example can be seen at this link.
(2) (PDF) Innovation in marketing strategy process: an integration and empirical examination (researchgate.net)
G-ECR-R is the Greek version of ECR-R. We have corrected it to the full name.
16.Table 5 – the heading of the attachment style is located at the left side for secure but for the other ones in the middle – in the tabeles you name the Beta-values as Beta, in the text as b not as ß
16.Response: We have moved the heading to the right place; for beta, we have corrected it to β.
17.P9 – it might be better understandable, if you provide a table of the correlations between the scales
17.Response: Yes, originally, we thought about that as well, but we were afraid that there would be too many tables. However, we have created the Table and put it in the supplement files.
(Table S1 in supplementary files).
18.Discussion: reference 34 is missing,
18.Response: We have added it.
- Oon-Arom A, Wongpakaran T, Kuntawong P, Wongpakaran N: Attachment anxiety, depression, and perceived social support: a moderated mediation model of suicide ideation among the elderly - ERRATUM. Int Psychogeriatr 2020, 32:1009.
19.P10 line 285 – reference is missing
19.Response: We have added it. (Bartholomew, 2021)
20.References – are presented in different citation styles - it should be consistently cited in one style
20.Response: We have corrected them.
Bartholomew, K. (2021). Self_Report Attachment Measures. Retrieved from Self-Report Attachment Measures - Kim Bartholomew - Simon Fraser University (sfu.ca)
Civilotti, C., Dennis, J. L., Acquadro Maran, D., & Margola, D. (2021). When Love Just Ends: An Investigation of the Relationship Between Dysfunctional Behaviors, Attachment Styles, Gender, and Education Shortly After a Relationship Dissolution. Frontiers in Psychology, 12, 2130.
Ostacoli, L., Cosma, S., Bevilacqua, F., Berchialla, P., Bovetti, M., Carosso, A. R., . . . Benedetto, C. (2020). Psychosocial factors associated with postpartum psychological distress during the Covid-19 pandemic: a cross-sectional study. BMC Pregnancy Childbirth, 20(1), 703. doi:10.1186/s12884-020-03399-5

Round 2
Reviewer 1 Report
I appreciate the authors' responses to my comments, and the changes made to the manuscript.
Regarding my point #6, I'm not sure if the authors are getting my point. I did not intend to attack the questionnaire method, but I think a better justification can be used. Yes interviews or objective tasks are time-consuming and impractical, but that do not automatically justify questionnaires if questionnaires are way off the mark. Thus, our assumption is that questionnaires are accurate "enough", given the time and financial constraints. This is why I mentioned how the authors can demonstrate that, yes, questionnaires are indeed accurate enough, by showing to the readers that 1) secure outcome is not abnormally high (thus showing no sign of social desirability) and that 2) correlation with depression and other symptoms is high (thus showing patients' ability to provide self-insight). These numbers, together, maybe with some other observations that I haven't noticed, can demonstrate the validity, usability, and practicality of questionnaires, and thus ameliorate the concern that I raised.
Author Response
25 Aug 2021
Dear editor
Thank you for your useful comments and suggestions. We have revised our manuscript as suggested (green texts). Please see below our point-by-point responses.
Comments and Suggestions for Authors
I appreciate the authors' responses to my comments, and the changes made to the manuscript.
Regarding my point #6, I'm not sure if the authors are getting my point. I did not intend to attack the questionnaire method, but I think a better justification can be used. Yes interviews or objective tasks are time-consuming and impractical, but that do not automatically justify questionnaires if questionnaires are way off the mark. Thus, our assumption is that questionnaires are accurate "enough", given the time and financial constraints. This is why I mentioned how the authors can demonstrate that, yes, questionnaires are indeed accurate enough, by showing to the readers that 1) secure outcome is not abnormally high (thus showing no sign of social desirability) and that 2) correlation with depression and other symptoms is high (thus showing patients' ability to provide self-insight). These numbers, together, maybe with some other observations that I haven't noticed, can demonstrate the validity, usability, and practicality of questionnaires, and thus ameliorate the concern that I raised.
Response: We are highly appreciated with the reviewer’s comments and suggestions without any negative feeling. We apologize if our message are unclear.
We revised this part to make the language clearer. Also, we have revised the conclusion as suggested.
Thank you very much.
We are looking forward to hearing from you.
Best regards,
Tinakon Wongpakaran
